# Contribution of Uremia to *Ureaplasma*-Induced Hyperammonemia

Derek Fleming,[a] Scott A. Cunningham,[a] Robin Patel[a,b]

[a]Division of Clinical Microbiology, Department of Laboratory Medicine and Pathology, Mayo Clinic, Rochester, Minnesota, USA
[b]Division of Infectious Diseases, Department of Medicine, Mayo Clinic, Rochester, Minnesota, USA

**ABSTRACT** Lung transplant recipients (LTRs) are vulnerable to hyperammonemia syndrome (HS) in the early postoperative period, a condition typically unresponsive to non-antibiotic interventions. HS in LTRs is strongly correlated with *Ureaplasma* infection of the respiratory tract, although it is not well understood what makes LTRs preferentially susceptible to HS compared to other immunocompromised hosts. *Ureaplasma* species harbor highly active ureases, and postoperative LTRs commonly experience uremia. We hypothesized that uremia could be a potentiating comorbidity, providing increased substrate for ureaplasmal ureases. Using a novel dialyzed flow system, the ammonia-producing capacities of four isolates of *Ureaplasma parvum* and six isolates of *Ureaplasma urealyticum* in media formulations relating to normal and uremic host conditions were tested. For all isolates, growth under simulated uremic conditions resulted in increased ammonia production over 24 h, despite similar endpoint bacterial quantities. Further, transcripts of *ureC* (from the ureaplasmal urease gene cluster) from *U. urealyticum* IDRL-10763 and ATCC-27816 rose at similar rates under uremic and nonuremic conditions, with similar endpoint populations under the two conditions (despite markedly increased ammonia concentrations under uremic conditions), indicating that the difference in ammonia production by these isolates is due to increased urease activity, not expression. Lastly, uremic mice infected with an *Escherichia coli* strain harboring a *U. urealyticum* serovar 8 gene cluster exhibited higher blood ammonia levels compared to nonuremic mice infected with the same strain. Taken together, these data show that *U. urealyticum* and *U. parvum* produce more ammonia under uremic conditions compared to nonuremic conditions. This implies that uremia is a plausible contributing factor to *Ureaplasma*-induced HS in LTRs.

**IMPORTANCE** *Ureaplasma*-induced hyperammonemia syndrome is a deadly complication affecting around 4% of lung transplant recipients and, to a lesser extent, other solid organ transplant patients. Understanding the underlying mechanisms will inform patient management, potentially decreasing mortality and morbidity. Here, it is shown that uremia is a plausible contributing factor to the pathophysiology of the condition.

**KEYWORDS** *Ureaplasma*, hyperammonemia, lung transplantation

Address correspondence to Robin Patel, patel.robin@mayo.edu.
The authors declare a conflict of interest. R.P. reports grants from ContraFect, TenNor Therapeutics Limited, Hylomorph, BioFire and Shionogi. R.P. is a consultant to Curetis, Specific Technologies, Next Gen Diagnostics, PathoQuest, Selux Diagnostics, 1928 Diagnostics, PhAST, Torus Biosystems, Mammoth Biosciences and Qvella; monies are paid to Mayo Clinic. R.P. is also a consultant to Netflix. In addition, R.P. has a patent on Bordetella pertussis/parapertussis PCR issued, a patent on a device/method for sonication with royalties paid by Samsung to Mayo Clinic, and a patent on an anti-biofilm substance issued. R.P. receives an editor's stipend from the Infectious Diseases Society of America (IDSA), and honoraria from the National Board of Medical Examiners (NBME), Up-to-Date and the Infectious Diseases Board Review Course. S.A.C. reports honorarium funds received from the Antibiotic Resistance Leadership Group.

Thousands of lung transplants are performed every year in the United States, with numbers anticipated to grow as availability and survivability continue to improve. A primary reason for greater survivability in recent years has been the incorporation of strategies to minimize mortality caused by posttransplant infections. A contributor that has plagued lung transplant recipient (LTR) survival is hyperammonemia syndrome (HS), which occurs in around 4% of LTRs (1, 2). Ammonia ($NH_3$) is a neurotoxin that, when present in excess, transverses the blood-brain barrier and causes cerebral edema (3, 4).

HS following lung transplantation typically progresses from early identification of elevated blood $NH_3$ levels or hyperammonemia (HA), causing altered mental status resulting in confusion, lethargy, obtundation, and agitation, to eventual cerebral

edema, resulting in seizure, coma, and often death (5–9). HS that presents in LTRs is atypical in that these patients do not have underlying liver disease or urea cycle disorders. Further, nontargeted interventional efforts to suppress endogenous $NH_3$ production biochemically or physiologically and/or increase $NH_3$ excretion have had minimal impact.

Recently, *Ureaplasma urealyticum* and *Ureaplasma parvum* were linked with HS in LTRs (10); the airways of every LTR presenting with unexplained HS studied ($n = 13$) tested positive for *Ureaplasma* species, likely of donor origin. Further evidence that *Ureaplasma* species are causative agents of HS in LTRs has come by way of *in vivo* studies with murine models, where it was shown that intratracheal combined with intraperitoneal infection with either *U. urealyticum* or *U. parvum* resulted in HA (11, 12). *Ureaplasma* species, which are normally considered commensal microbiota of the urogenital tract, produce a potent urease that splits urea into $NH_3$ and $CO_2$ as a means of ATP synthesis, powered by the $NH_3$ gradient generated across the membrane (13, 14). Interestingly, 95% of ATP generated by *Ureaplasma* species is urea dependent, making it a requirement for growth (15). The high level of $NH_3$ production from LTR *Ureaplasma* infection can exceed the capacity for detoxification by the host.

While unexplained HS has been described in non-LTR transplant patient populations (2, 16–30), the prevalence rate of ~4% seems highest, by far, in the LTR population. This suggests that the lung transplant scenario is specifically well suited to microbe-driven HS. LTRs are also particularly vulnerable to posttransplant uremia (31, 32), potentially providing an abundance of substrate for ureaplasmal ureases, leading to $NH_3$ overproduction, overwhelming the detoxification capacity of the host.

Here, the possibility that elevated blood urea in LTRs could potentially contribute to production of pathological levels of $NH_3$ by *Ureaplasma* species was investigated. We hypothesized that, compared to nonuremic conditions, uremic conditions would result in increased $NH_3$ production by ureaplasmal ureases both *in vitro* and *in vivo*, ultimately leading to a greater incidence of HS.

## RESULTS

**All *Ureaplasma* isolates produced significantly more ammonia under conditions related to uremia.** For all *U. parvum* and *U. urealyticum* isolates tested, growth in media containing 50 mg/dL urea (uremic conditions) resulted in significantly greater $NH_3$ production than in media containing 10 mg/dL urea (normal conditions) over 24 h (Fig. 1). Isolates grown under uremic conditions produced, on average, 1777 (standard deviation [SD] = 264) $\mu$mol/L more $NH_3$ than those grown under normal conditions. $NH_3$

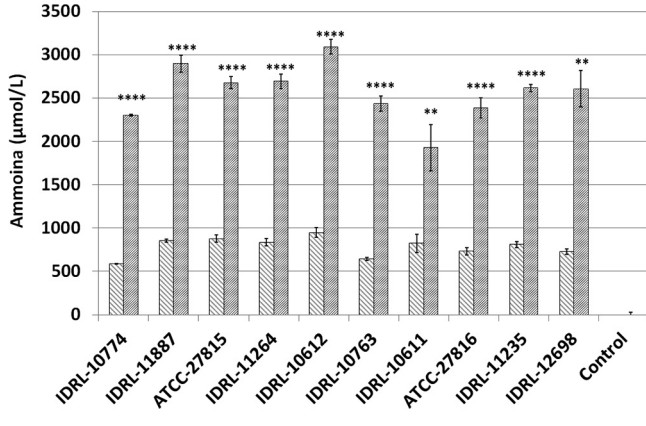

**■ 10 mg/dL Urea ■ 50 mg/dL Urea**

**FIG 1** Average ammonia production by *Ureaplasma* isolates over 24 h under normal and uremic conditions. Isolates of *U. parvum* or *U. urealyticum* were grown in the dialyzed flow system under normal (10 mg/dL urea) or uremic (50 mg/dL urea) conditions for 24 h. The experiments were performed in triplicate. The control was uninfected medium. The significance between conditions for each isolate was determined via two-tailed unpaired *t* tests. **, $P \leq 0.01$; ****, $P \leq 0.0001$.

production was not significantly different between species, with *U. parvum* and *U. urealyticum* isolates producing averages of 1,854 $\mu$mol/L (SD = 67) $\mu$mol/L and 1,731 $\mu$mol/L (SD = 131) more $NH_3$, respectively, under uremic conditions. There was no noticeable difference between patient respiratory isolates and commercially available urogenital isolates. The average difference in $NH_3$ production between uremic and normal conditions for all patient respiratory isolates was 1,794 $\mu$mol/L (SD = 291) *versus* 1801 $\mu$mol/L (SD = 72) and 1,655 $\mu$mol/L (SD = 119) for *U. parvum* (ATCC 27815) and *U. urealyticum* (ATCC 27816) urogenital isolates, respectively. Between uremic and normal conditions, 24-h color changing units (CCU) counts were not significantly different (Fig. S2), indicating that $NH_3$ production was not a result of different numbers of bacteria.

**Increased ammonia production under uremic conditions was not due to increased urease production in two isolates of *U. urealyticum*.** Quantitative PCR (qPCR) was used to measure *ureC* gene copies, and quantitative reverse transcription PCR (qRT-PCR) to measure *ureC* transcripts over time for one clinical respiratory isolate (IDRL-10763) and one commercially available urogenital isolate (ATCC-27816) of *U. urealyticum*. It was found that, despite a significant increase in $NH_3$ production for both isolates in the 50 mg/dL urea preparation, and despite similar endpoint populations (IDRL-10763: $7.23 \times 10^8$ and $7.84 \times 10^8$ copies/mL; ATCC-27816: $9.24 \times 10^8$ and $7.84 \times 10^8$ copies/mL for 10 and 50 mg/dL urea, respectively), rises in *ureC* transcripts were similar across conditions (Fig. 2). This indicates that greater urea availability does not result in increased urease production by *U. urealyticum*.

**Uremic mice infected with an *Escherichia coli* strain expressing the *U. urealyticum* gene cluster exhibited elevated blood ammonia levels compared to nonuremic mice.** It was determined whether an *E. coli* strain harboring the *U. urealyticum* gene cluster (33) would produce more $NH_3$ in uremic compared to nonuremic mice. Mice were immunosuppressed for 7 days with methylprednisone, tacrolimus, and mycophenolate mofetil, and half of the animals were administered 40 g/L urea *ad libitum* in their drinking water beginning 10 days prior to infection, resulting in elevated blood urea nitrogen (BUN) levels for urea-fed mice in comparison to control mice. The animals were infected intratracheally (IT) and intraperitoneally (IP) with $10^7$ to $10^8$ cells/mL of the urease-positive strain, the wild-type control, or saline + 0.1% agar (vehicle control). After 24 h, the animals were sacrificed; their blood was collected for $NH_3$ measurement with a point-of-care meter (Woodley Equipment Company Ltd., WD5502 PocketChem

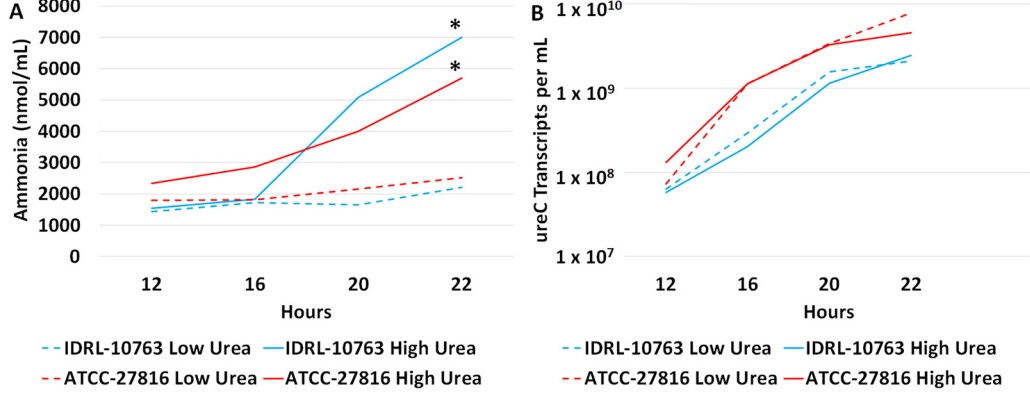

**FIG 2** *ureC* transcript levels are similar with normal and elevated urea availability. $10^5$ CFU/mL of *U. urealyticum* IDRL-10763 or *U. urealyticum* ATCC-27816 was grown in 10 mL 100 mM 2-(*N*-morpholino)ethanesulfonic acid (MES)-buffered U9 for 2 h at 37°C and added to a 1,000-kDa-pore-size dialysis tube that was submerged in a 250-mL flow bottle containing 100 mM MES-buffered U9 with either 10 or 50 mg/dL urea with a flow rate of 2 mL/h. (A) $NH_3$ was measured via the modified Berthelot reaction at each time point from the cell-free exterior of the dialysis tube. (B) Quantitative reverse transcription PCR (qRT-PCR) was performed on 50 $\mu$L of culture collected at 12, 16, 20, and 22 h and placed in DNA/RNA shield. DNA was extracted using the Maxwell rapid sample concentrator, with a standard $10^8$ copies/mL extraction control from the same stock of culture in DNA/RNA shield run for each extraction group. qRT-PCR was performed on a Roche LightCycler 2.0 with crossing threshold values compared to a standard curve. The significance between urea concentrations was determined via linear regression analysis: *, $P \leq 0.05$.

BA Analyzer), blood CFU measurement (dilution and plating), and *U. urealyticum ureC* qPCR; and their lungs were harvested for total lung CFU measurement and *U. urealyticum ureC* qPCR. Despite insignificant differences between lung and blood populations (Fig. S3), infection with urease-positive *E. coli* in urea-fed mice resulted in significantly higher blood $NH_3$ levels than all other groups (Fig. 3).

## DISCUSSION

The results of this study show that conditions representative of uremia resulted in elevated production of $NH_3$ by all *Ureaplasma* isolates tested. The discovery that *Ureaplasma* respiratory infections were the cause of the previously unexplained phenomenon of nonhepatic HS in early postoperative LTRs (10) has led to an improvement in patient care and a reduction in mortality rates. Still, what makes LTRs particularly vulnerable to this phenomenon remains unknown. Here, the potential impact of uremia in the early post-transplant period as a potentiating comorbidity was investigated. LTRs are frequently uremic due to acute renal failure (ARF). It has been estimated that as many as 75% of LTRs experience ARF postoperation due to renal hypoperfusion brought on by several factors, including (i) decreased circulating blood volume resulting from diuretic use to prevent pulmonary edema from leaky capillaries, (ii) nephrotoxic effects of calcineurin inhibitors, and/or (iii) reduced renal oxygenation due to postoperation hypoxia (31–36). We hypothesized that, among patients harboring a posttransplant *Ureaplasma* respiratory infection, elevated blood urea concentrations would provide greater substrate availability for ureaplasmal ureases, leading to the production of $NH_3$ at sufficient levels to overwhelm the host detoxification capacity (Fig. 4).

Using a novel dialyzed flow system, $NH_3$ production of 10 isolates of *U. parvum* and *U. urealyticum* (8 clinical respiratory isolates and 2 commercially available urogenital isolates) over 24 h under conditions representative of normal (10 mg/dL) and elevated (50 mg/dL) BUN was tested. For all isolates tested, more $NH_3$ was produced under uremic conditions. Further, endpoint population densities were not significantly different for normal compared to uremic conditions, indicating that greater $NH_3$ production under uremic conditions can be credited to increased ureaplasmal urease activity. Previous studies have shown that *Ureaplasma* growth capacity is highly dependent on

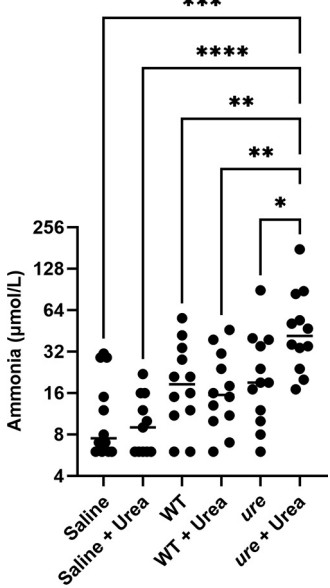

**FIG 3** Blood ammonia is elevated in uremic mice infected with urease-positive *E. coli* compared to nonuremic mice infected with the same strain. Urea-fed (+ Urea) and normal mice were infected intraperitoneally and intratracheally with either wild-type (WT) or urease-positive *E. coli* (*ure*) or 0.1% saline agar. Blood $NH_3$ was measured 24 h later ($N$ = 12 animals per group; one-way analysis of variance [ANOVA]): *, $P \leq 0.05$; **, $P \leq 0.01$; ***, $P \leq 0.001$; ****, $P \leq 0.0001$.

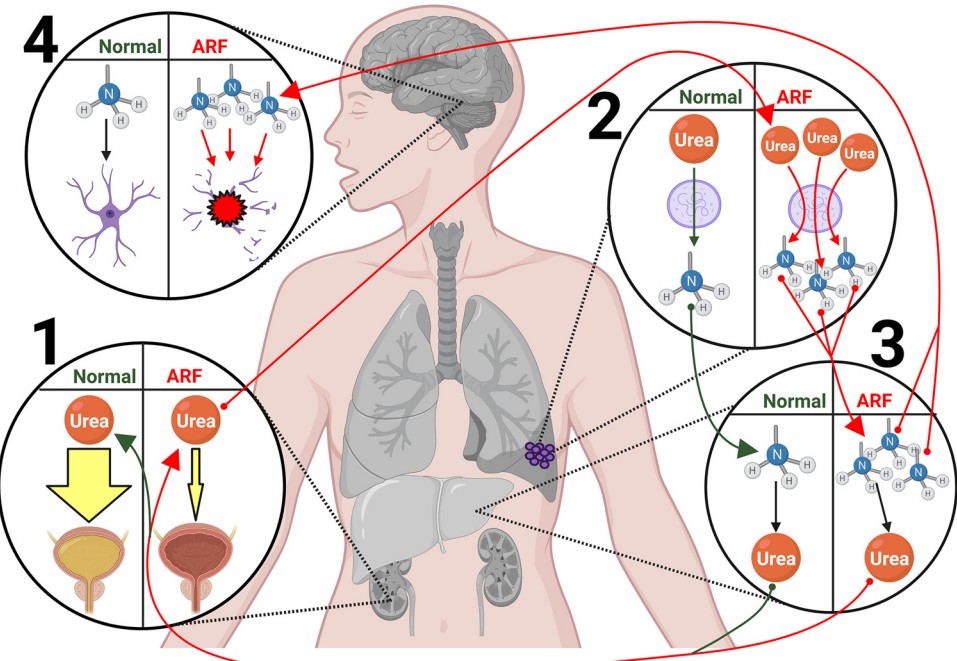

**FIG 4** Hypothetical mechanism of *Ureaplasma*-induced hyperammonemia syndrome in the context of kidney dysfunction in early post-transplant lung transplant recipients. Circle 1 shows that acute renal failure (ARF) due to renal hypoperfusion results in decreased elimination of urea from the blood via urine excretion. Circle 2 shows that increased blood urea availability provides additional substrate for ureaplasmal (purple cells) ureases in the infected respiratory tract, leading to greater $NH_3$ (blue and gray $NH_3$ molecule) production in comparison to normal blood urea concentrations. Circle 3 shows that elevated serum $NH_3$ levels overwhelm the detoxification capacity of liver urea cycle enzymes, leading to hyperammonemia. Circle 4 shows that excess serum $NH_3$ diffuses into astrocytes and other glial cells in the brain, causing them to swell and burst, resulting in cerebral edema and hyperammonemia syndrome. The figure was created using BioRender.

urea concentration *in vitro* (37). However, the novel flow system utilized here was designed to supply continuous urea at the desired concentration while at the same limiting increases in alkalinity to growth-inhibitory levels. With the same system, a comparison of $NH_3$ production and qPCR and qRT-PCR copy counts by two isolates of *U. urealyticum* showed that the increase in $NH_3$ production under the elevated urea condition over time was not the result of elevated urease production or greater bacterial population densities in comparison to the lower urea condition.

With this *in vitro* data in hand, an *E. coli* strain harboring the *U. urealyticum* gene cluster was used to test whether uremic mice would experience greater ureaplasmal urease-induced HA compared to nonuremic mice. By supplying animals with urea in drinking water, mildly elevated BUN was achieved in comparison to mice given normal water. Uremic mice infected with the ureaplasmal urease-positive strain of *E. coli* experienced elevated blood $NH_3$ levels in comparison to nonuremic mice infected with ureaplasmal urease-positive *E. coli*, as well as all other groups, including uremic and nonuremic animals infected with either urease-negative *E. coli* or vehicle control.

There are several limitations to this study. First, these results were obtained with conditions related to mild, dietary uremia. More significant differences in BUN could be obtained with models of acute kidney injury, replicating conditions of ARF, or with non-*ad libitum* methods of urea feeding, which would theoretically lead to more significant differences in ureaplasmal urease-induced HA, given the enzyme capacities demonstrated herein (Fig. 4). Second, in the *in vivo* portion of this study, only the serotype 8 urease gene cluster was investigated; it is possible that other serotypes would not have the same level of responsiveness to increased urea concentrations in the blood. To this end, further investigation will be needed to determine universality across *Ureaplasma* species. Further, *in vivo* studies were performed using an *E. coli* strain that harbors a plasmid containing the *U. urealyticum* gene

cluster and not a *Ureaplasma* isolate. This was done to provide a non-urease-producing control and because *E. coli* is easier to work with experimentally than are *Ureaplasma* species. Lastly, we chose to perform a qPCR/qRT-PCR comparison study (Fig. 2) on two representative *U. urealyticum* isolates from Fig. 1. Although the studied isolates showed increased $NH_3$ production under uremic conditions, it is possible that urease expression may manifest differently in certain isolates.

Taken together, these results provide strong *in vitro* and *in vivo* support for the hypothesis that posttransplant uremia may be a contributing factor to *Ureaplasma*-induced HS in LTRs. This study serves as a reminder that the effects of specific comorbidities on diseases of microbial metabolite production/overproduction should be more readily considered.

## MATERIALS AND METHODS

**Study isolates.** The isolates of *U. parvum* and *U. urealyticum* investigated here are listed in Table 1. They include three respiratory isolates of *U. parvum* and five respiratory isolates of *U. urealyticum*, as well as one commercially available urogenital isolate of each species (from ATCC). Patient respiratory isolates are stored at the Mayo Clinic Infectious Diseases Research Laboratory (IDRL). The isolates were grown to $10^7$ CCU using a *Ureaplasma* bioreactor, as previously described (38). Aliquots in U9 media (Hardy Diagnostics) buffered with 100 mM 2-(*N*-morpholino)ethanesulfonic acid (MES) at pH 6.0, 500-$\mu$L aliquots were frozen at $-80°C$ until use.

An opal suppressor (under the control of an isopropylthio-$\beta$-D-galactoside [IPTG]-induced lacUV5 promoter) strain of *E. coli* harboring a plasmid containing the urease gene cluster of serotype 8 *U. urealyticum* (33) was utilized to study $NH_3$ production in uremic *versus* nonuremic mice. The plasmid carried both chloramphenicol and ampicillin resistance for selection and maintenance of urease activity. The wild-type strain lacked the plasmid and was thus urease-negative and susceptible to both antibiotics. The cultures were grown in tryptic soy broth (TSB) for 18 h, with or without antibiotics, at 37°C with shaking, after which IPTG was added, and the cultures were incubated an additional 4 h. Prior to infection, urease production by the *E. coli* strain was verified by streaking on urea agar slants (Hardy Diagnostics; R42).

**Dialyzed flow system.** 10-mL cultures of $10^5$ CCU/mL for all *Ureaplasma* isolates were encased in dialysis tubing (Specta/Por Float-A-Lyzer G2 1,000-kDa dialysis device; G235073) and submerged in 250 mL of 100 mM MES-buffered U9, allowing measurement of $NH_3$ levels over time (Fig. 5). The entire device was incubated at 37°C, with fresh broth added and spent media removed via flow at 2 mL/hour. Urea concentrations in the growth media were varied to mimic normal and high BUN levels (10 and 50 mg/dL, respectively) in the flow chamber and inflow and were maintained at a standard deviation of 3 for the 50 mg/mL concentration and 1 for the 10 mg/mL concentration. The samples were taken from cell-free portions of the flow chamber (outside the dialysis tube) and collected into microcentrifuge tubes at 0 and 24 h, and $NH_3$ and urea concentrations were tested using an $NH_3$ assay kit (Abcam; ab102509) and a urea assay kit (Abcam; ab234052). *Ureaplasma* cells in the dialysis tubing were quantified at each collection time.

For qPCR and qRT-PCR measurements over time, $10^5$ CFU of *U. urealyticum* IDRL-10763 or ATCC-27816 was inoculated into 100 mM MES-buffered U9 medium and grown for 2 h at 37°C. After 2 h, the cultures were submerged in 250 mL of 100 mM MES-buffered U9 medium containing either 10 or 50 mg/dL urea and grown at 37°C for 22 h with 2 mL/h inflow and outflow from the exterior of the dialysis tube (i.e., cell-free portion of the system). At hours 12, 16, 20, and 22, 150 $\mu$L of culture material was collected from the dialysis tube, mixed 1:1 in DNA/RNA shield (Zymo Research), and stored at $-80°C$ until quantitative PCR (qPCR) and quantitative reverse transcriptase PCR (qRT-PCR) assay. Additionally, 200 $\mu$L of medium from the exterior of the dialysis tube was collected and stored at $-80°C$ for $NH_3$ assay (Abcam, modified Berthelot) and urea assay. Sampling volumes at each time point (which were the same across all experimental groups) were not replaced as each draw represented 0.13% of the total volume per time point (which would be expected to have an insignificant impact on $NH_3$ and urea quantification).

**TABLE 1** *Ureaplasma* isolates studied

| Species | Isolate no. | Source |
| --- | --- | --- |
| *U. parvum* | IDRL-10774 | Bronchoalveolar lavage fluid |
| *U. parvum* | IDRL-11887 | Bronchoalveolar lavage fluid |
| *U. parvum* | IDRL-11264 | Sputum |
| *U. parvum* | ATCC-27815 | Urethritis, serovar 3 |
| *U. urealyticum* | IDRL-10763 | Bronchial washings |
| *U. urealyticum* | IDRL-10612 | Bronchoalveolar lavage fluid |
| *U. urealyticum* | IDRL-10611 | Bronchoalveolar lavage fluid |
| *U. urealyticum* | IDRL-11235 | Tracheal secretions |
| *U. urealyticum* | IDRL-12698 | Bronchoalveolar lavage fluid |
| *U. urealyticum* | ATCC-27816 | Urethritis, serovar 4 |

Microbiology
Spectrum

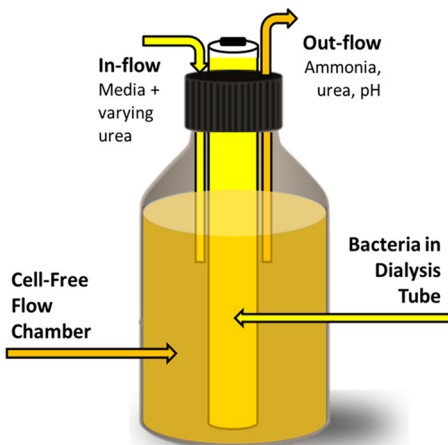

**FIG 5** Schematic of the dialyzed flow system.

**Experimental mouse model.** C3H male and female mice (18 to 22 g; Charles River Laboratories, Wilmington, MA) were pharmacologically immunosuppressed for 7 days with methylprednisone, tacrolimus, and mycophenolate mofetil; half of the animals were administered 40 g/L urea *ad libitum* in their drinking water beginning 10 days prior to infection, resulting in an average blood urea nitrogen level (BUN) of 38 mg/dL (SD = 14) for urea-fed mice and 28 mg/dL (SD = 6.97) for control mice (unpaired *t* test: *P* = 0.0156). For IT challenge, the mice were anesthetized with ketamine/xylazine (90/10 mg/kg); 50 $\mu$L of bacterial suspension (selected to ensure ureaplasmal entry into the airways as used in our prior work [11, 12]) was placed into their trachea using a 22-gauge curved gavage needle, after which the animals were placed in a vertical position for 10 min using a murine vertical stabilization apparatus (Fig. S1). For IP challenge, 100 $\mu$L of $10^7$ to $10^8$ CFU/mL bacterial suspension was injected into the peritoneum. After 24 h of infection, the animals were sacrificed, their blood was collected for $NH_3$ measurement with a point-of-care meter (Woodley Equipment Company Ltd., WD5502 PocketChem BA analyzer), blood CFU measurement (dilution and plating), and *U. urealyticum ureC* qPCR, and their lungs were harvested for total lung CFU measurement and *U. urealyticum ureC* qPCR. A total of 12 mice were assigned to each experimental group. A sample size of 12 mice per group yields 80% power to detect a difference of 1.75 standard deviations or larger for levels of $NH_3$ between two groups using two sample *t* tests.

**Ethics statement.** This study was carried out in accordance with the recommendations in the Guide for the Care and Use of Laboratory Animals of the National Institutes of Health and was approved by Mayo Clinic Institutional Animal Care and Use Committee (protocol number A5004-20). The Mayo Clinic is Association for Assessment and Accreditation of Laboratory Animal Care (AAALAC) accredited (000717), registered with the U.S. Department of Agriculture (41-R-0006), and has an Assurance with the Office of Laboratory Animal Welfare (A3291-01). The mice were housed in a biosafety level 2, specific-pathogen-free, AAALAC-accredited facility. Sentinel mice were tested quarterly for murine pathogens; all were negative throughout the course of this study. The mice had *ad libitum* access to irradiated rodent food (LabDiet formula 5053) and water. The facility was environmentally controlled (temperature, 68 to 74°F; relative humidity, 30 to 70%; 12:12-h light:dark cycle). All efforts were made to minimize suffering. The animals were monitored twice daily, and anesthetized mice were monitored until awake. Mice were monitored for decreased activity, decreased body temperature, hunched stature, distress, and inability to eat and drink; if these occurred and were severe, the animals were humanely euthanized.

**qPCR and qRT-PCR assays.** Nucleic acids were purified from tissue and culture material using a Maxwell RSC (Promega, Madison WI). DNA isolation was carried out with a Maxwell RSC tissue DNA kit and RNA isolated with a Maxwell RSC simplyRNA tissue kit as per the manufacturer's instructions. Sample input for both kits was 100 $\mu$L, and elution output was 100 $\mu$L for DNA and 50 $\mu$L for RNA.

The quantitative PCR assay for *U. parvum* and *U. urealyticum* has been previously described (39) and was performed with slight modifications to adapt the assay for use on the LightCycler 2.0 instrument. Briefly, 1× LightCycler DNA Master HybProbe was combined with 1× primer/probe set 1408 (TIB Molbiol, Howell Township, NJ) and an additional 3 mM $MgCl_2$. A total of 15 $\mu$L of the complete master mix was combined with 5 $\mu$L of DNA extract in a LightCycler sample capillary. Thermocycling conditions were as follows: Denaturation at 95°C for 10 min; amplification for 45 cycles of 10 s at 95°C, 15 s at 60°C (single acquisition), and 15 s at 72°C; melting curve analysis/amplicon detection for 0 s at 95°C, 20s at 59°C, 20 s at 40°C (ramp rate of 0.2°C/s), 0 s at 80°C (ramp rate of 0.2°C/s and continuous acquisition); and cooling for 30 s at 40°C.

The qRT-PCR assay was adapted from the above qPCR assay to include a RT-specific reverse primer (5′-TTGNTCAAANATTGGATCTTCC-3′). The complete master mix included 1× primer/probe set 1408, 0.25 mM RT-specific reverse primer, 1× SuperScript III Platinum One-Set qRT-PCR system buffer, 0.05× SuperScript enzyme, and 1 mM $McCl_2$ (Invitrogen, Waltham, MA). 15 $\mu$L of the complete RT master mix was combined with 5 $\mu$L of RNA extract in a LightCycler sample capillary. The thermocycling conditions were as follows: reverse transcription/denaturation at 55°C for 15 min followed by 95°C for 2 min; amplification for 45 cycles of 10 s at 95°C, 20 s at 60°C (single acquisition), and 20 s at 72°C; and cooling for

30 s at 40°C. Plasmid control material (30-8706-01 *U. urealyticum* and 30–8706-03 *U. parvum*; TIB Molbiol) were serially diluted and used to generate quantification lines for each species. A single, $1 \times 10^3$ midpoint dilution of was included as a calibration control. Prequantified culture material preserved in DNA/RNA shield was included as a process control and carried through extraction and assayed with each experiment.

## SUPPLEMENTAL MATERIAL

Supplemental material is available online only.

**SUPPLEMENTAL FILE 1**, PDF file, 0.1 MB.

## ACKNOWLEDGMENTS

We thank Melissa Karau for her assistance with this project. We also thank the Clinical Microbiology Laboratory at the Mayo Clinic (Rochester, Minnesota) for providing the clinical isolates used in this study and Alain Blanchard from the University of Bordeaux for providing the urease-positive *E. coli* strain.

R.P. reports grants from ContraFect, TenNor Therapeutics Limited, Hylomorph, BioFire, and Shionogi. R.P. is a consultant to Curetis, Specific Technologies, Next Gen Diagnostics, PathoQuest, Selux Diagnostics, 1928 Diagnostics, PhAST, Torus Biosystems, Mammoth Biosciences, and Qvella; monies are paid to the Mayo Clinic. R.P. is also a consultant to Netflix. In addition, R.P. has a patent on *Bordetella pertussis/Bordetella parapertussis* PCR issued, a patent on a device/method for sonication with royalties paid by Samsung to the Mayo Clinic, and a patent on an anti-biofilm substance issued. R.P. receives an editor's stipend from the Infectious Diseases Society of America (IDSA) and honoraria from the National Board of Medical Examiners (NBME), Up-to-Date, and the Infectious Diseases Board Review Course. S.A.C. reports honorarium funds received from the Antibiotic Resistance Leadership Group.

The research reported in this publication was supported by the National Institute of Allergy and Infectious Diseases of the National Institutes of Health under award number R21AI150649. The content is solely the responsibility of the authors and does not necessarily represent the official views of the National Institutes of Health.

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
