## [Reviewer comments · Microbiology Spectrum]

Microbiology Spectrum

Contribution of Uremia to *Ureaplasma*-Induced Hyperammonemia

Derek Fleming, Scott Cunningham, and Robin Patel

Corresponding Author(s): Robin Patel, Mayo Clinic

Review Timeline:

Submission Date:	October 19, 2021
Editorial Decision:	November 16, 2021
Revision Received:	January 14, 2022
Accepted:	January 16, 2022

Editor: Karen Carroll

Reviewer(s): The reviewers have opted to remain anonymous.

Transaction Report:

DOI: <https://doi.org/10.1128/spectrum.01942-21>

November 16, 2021

Dr. Robin Patel
Mayo Clinic
200 First Street SW
Rochester, Minnesota 55905

Re: Spectrum01942-21 (Contribution of Uremia to *Ureaplasma*-Induced Hyperammonemia)

Dear Dr. Robin Patel:

Thank you for submitting your manuscript to Microbiology Spectrum. Two reviewers have submitted their comments with suggestions for significant revisions with which I agree. Both reviewers are concerned that the results of the manuscript do not support some of the claims made in the discussion. Both reviewers would like to see more transparent comments regarding limitations of the work. That said, both reviewers find value in the experiments. Therefore I am inviting a resubmission of a revised manuscript that addresses in detail both reviewers' concerns.

Link Not Available

Sincerely,

Karen Carroll

Journals Department
Reviewer comments:

Reviewer #1 (Comments for the Author):

In this manuscript the authors present an in vitro system of measuring urea metabolism by commercial and clinically acquired *Ureaplasma* species, as well as an in vivo murine model of hyperammonemia driven by *E. coli* expressing *Ureaplasma* urealyticum urease enzymes. An ethics statement is provided for the animal studies, and all experiments include appropriate controls.

The primary strength of this study is development of a dialyzed flow system that allows monitoring of metabolic processes from cultured *Ureaplasma* species in real-time. This is a useful system with great potential for investigating the metabolic capacity of *Ureaplasma* spp. However, this study lacks robust data evaluating the system. Although the in vitro experiments presented in the manuscript are well designed and well controlled, the data presented in Figure 5 and Supplemental Table 1 do not present results from enough *Ureaplasma* isolates to support the conclusions made in the manuscript. Data from only a single *Ureaplasma* urealyticum clinical isolate (IDRL-10763) are presented to investigate levels of ureC gene abundance / transcription

and ammonia production at multiple timepoints, whereas ten isolates (eight clinical and two commercial) are investigated for the overall change in ammonia production (Figure 4) at the 24 hour timepoint. This reviewer is not convinced that the isolate selected adequately represents the other nine isolates presented in Figure 4, which may demonstrate species-level differences or differences between clinically isolated and commercially passaged strains. Additional studies utilizing all strains (ideally) or representative strains (at the minimum) evaluating the gene copy number, transcript abundance, and production of ammonia are needed to support the claims made in this manuscript. This reviewer does not think that q1 hour timepoints are necessary (as presented in Figure 5), but studies with additional isolates and reasonably spaced timepoints are needed.

An additional strength of this manuscript is the utilization of an in vivo model system to evaluate the contribution of the U. urealyticum serotype 8 urease gene cluster in development of hyperammonemia. The in vivo studies are well designed and well controlled, and it is commendable that the authors provide in vivo data in association with the in vitro data. However, conclusions made from this model are too far reaching, and there are significant limitations to this model which are not adequately addressed, as follows:

- 1) The most significant concern is framing this as a model of uremia due to acute renal failure. This model is not based upon acute renal failure but rather excess dietary consumption of urea. This is an acceptable model for uremia in-and-of-itself, however, this model does not incorporate other variables present in the context of acute renal failure that could influence urea metabolism by Ureaplasma species and the subsequent development of hyperammonemia. Thus, the conclusions made about hyperammonemia resulting from Ureaplasma species metabolizing urea specifically in the setting of acute renal failure are not supported by the data presented (ie lines 43-44, 234-235, and 271-272).
- 2) A second concern regards extrapolation of conclusions made from this model. For example, in the abstract it is stated that these data "show that U. urealyticum and U. parvum produce more ammonia under uremic conditions ... in vivo". However, this statement cannot be supported by the data presented as the in vivo data only investigated a model of U. urealyticum serotype 8. If the urease gene cluster from this specific Ureaplasma species and serotype is identical to that of other species and serotypes and thus is a model for all Ureaplasma species, this should be explained for clarity. If not, then limitations of this model must be expounded upon further.
- 3) A final concern regards the use of E. coli expressing Ureaplasma urease gene clusters in the model. This is an adequate model to isolate the contribution of the specific urease gene cluster expressed on this plasmid. However, no comment is made as to why an artificial expression model was used rather than live U. urealyticum and/or U. parvum (as was used in references 11 and 12). This needs to be elucidated in the manuscript, with limitations of this model clearly explained.

Additional comments are as follows:

- The schematics presented in the introduction (Figures 1 and 2) seem better suited for a review of hyperammonemia. They are well designed and visually impressive, but the specific mechanisms presented (particularly those in Figure 1 involving ammonia and glutamate metabolism in astrocytes and neurons) lead the reader to think these will be experimentally targeted in the body of the work. They are not, and the reader is left wondering why such detail was included that does not contextualize the specific experiments presented.
- A hypothesis is presented in the abstract and the discussion, but not in the introduction. Please include hypotheses in the introduction to provide the reader with an understanding of the questions addressed in the study.
- Even when non-significant p values are present, please include the p values in the text rather than stating that compared values are "not significantly different".
- Data reporting the BUN of experimental animals is repeated three times in the manuscript (lines 123-124, 217-218, and 262). It is sufficient to only report this in the results and then comment in the discussion.
- There are minor typos throughout the manuscript, the most significant being the title subheading on line 211-212 describes an "E. coli strain expressing U. urealyticum". It seems the term "urease gene cluster" was omitted.

Reviewer #2 (Comments for the Author):

This is an interesting study which utilises in vivo and in vitro animal models to suggest that ureaplasmas may be a cause of hyperammonemia syndrome in lung transplant recipients. I have several comments and suggestions that would improve the quality of the manuscript overall:

- Please clarify why you are using CCU as a quantification measure, instead of CFU. CFU is by far the more accurate measure of growth, so it would be useful to provide a rationale for CCU which of course can be used, but is not as accurate.
- Lines 89-96: did you assess ammonia production from the urease-containing plasmid prior to inoculation into animals?
- With regards to your animal challenge methodology - 50 uL seems like a large volume for intratracheal challenge. I suspect this would overflow and some of the volume would go into the stomach. Can you please comment on the volume used here as I think a smaller volume would have been better for this particular challenge route in the mice, to ensure the total volume remained in the required site. Similarly, can you please clarify for each experiment how many mice were used for each experimental group and the statistical tests used to inform the appropriate experimental group size.
- in your dialysed flow system, I can see that media is added and spent media removed at 2 ml per hour. Did you also take into

account the need for additional volumes of media to replace the volumes of media being taken for sampling (or was the spent media used to assess the urea/ammonia concentrations? Some clarification on this point would be good as changes in the total volume of the system can affect the concentrations of some of these substrates.

- For your nucleic acid extraction, can you please clarify if you have used the manufacturers instructions or if there was any deviation from the standard workflow/instructions.

- For figure 4, please change the black bars (or the bottom error bar) so you can clearly see the variation in results. Also note in the legend for this figure, you currently have written " $p \leq 0.01$, " $p \leq 0.0001$ " and this needs to be amended (presumably the second value is meant to be for ****?)

- I think you could be a little more transparent with the limitations of this work and this could be addressed with a few additional sentences added to the discussion.

Staff Comments:

Preparing Revision Guidelines

Please return the manuscript within 60 days; if you cannot complete the modification within this time period, please contact me. If you do not wish to modify the manuscript and prefer to submit it to another journal, please notify me of your decision immediately so that the manuscript may be formally withdrawn from consideration by Microbiology Spectrum.

Response to Reviewers

Reviewer #1 (Comments for the Author):

In this manuscript the authors present an in vitro system of measuring urea metabolism by commercial and clinically acquired *Ureaplasma* species, as well as an in vivo murine model of hyperammonemia driven by *E. coli* expressing *Ureaplasma urealyticum* urease enzymes. An ethics statement is provided for the animal studies, and all experiments include appropriate controls.

The primary strength of this study is development of a dialyzed flow system that allows monitoring of metabolic processes from cultured *Ureaplasma* species in real-time. This is a useful system with great potential for investigating the metabolic capacity of *Ureaplasma* spp. However, this study lacks robust data evaluating the system. Although the in vitro experiments presented in the manuscript are well designed and well controlled, the data presented in Figure 5 and Supplemental Table 1 do not present results from enough *Ureaplasma* isolates to support the conclusions made in the manuscript. Data from only a single *Ureaplasma urealyticum* clinical isolate (IDRL-10763) are presented to investigate levels of ureC gene abundance / transcription and ammonia production at multiple timepoints, whereas ten isolates (eight clinical and two commercial) are investigated for the overall change in ammonia production (Figure 4) at the 24 hour timepoint. This reviewer is not convinced that the isolate selected adequately represents the other nine isolates presented in Figure 4, which may demonstrate species-level differences or differences between clinically isolated and commercially passaged strains. Additional studies utilizing all strains (ideally) or representative strains (at the minimum) evaluating the gene copy number, transcript abundance, and production of ammonia are needed to support the claims made in this manuscript. This reviewer does not think that q1 hour timepoints are necessary (as presented in Figure 5), but studies with additional isolates and reasonably spaced timepoints are needed.

We thank the reviewer for their enthusiasm about our work, for their time spent analyzing the manuscript, and for their valuable input. We agree that one isolate was not enough to draw broad conclusions from figure 5 (now figure 3). Due to the labor and time intensiveness of the protocol, we chose to analyze an additional representative isolate of *U. urealyticum*, particularly a commercially available (ATCC strain) strain, at timepoints of 12, 16, 20, and 22 hours. With this new data, we reworked the figure and made appropriate changes to the text to indicate the inclusion of an additional isolate. Additionally, we added text to the Discussion to acknowledge the limitation of not testing every isolate in this manner.

An additional strength of this manuscript is the utilization of an in vivo model system to evaluate the contribution of the *U. urealyticum* serotype 8 urease gene cluster in development of hyperammonemia. The in vivo studies are well designed and well controlled, and it is commendable that the authors provide in vivo data in association with the in vitro data. However, conclusions made from this model are too far reaching, and there are significant limitations to this model which are not adequately addressed, as follows:

- 1) The most significant concern is framing this as a model of uremia due to acute renal failure. This model is not based upon acute renal failure but rather excess dietary consumption of urea. This is an acceptable model for uremia in-and-of-itself, however, this model does not incorporate

other variables present in the context of acute renal failure that could influence urea metabolism by *Ureaplasma* species and the subsequent development of hyperammonemia. Thus, the conclusions made about hyperammonemia resulting from *Ureaplasma* species metabolizing urea specifically in the setting of acute renal failure are not supported by the data presented (ie lines 43-44, 234-235, and 271-272).

We appreciate this feedback and agree that we don't have enough evidence to link this specifically to acute renal failure. As such, we have removed any direct mention of acute renal failure, and instead mentioned that uremia could be caused by acute renal failure, and that the contribution of acute renal failure should be further investigated in the future.

2) A second concern regards extrapolation of conclusions made from this model. For example, in the abstract it is stated that these data "show that *U. urealyticum* and *U. parvum* produce more ammonia under uremic conditions ... *in vivo*". However, this statement cannot be supported by the data presented as the *in vivo* data only investigated a model of *U. urealyticum* serotype 8. If the urease gene cluster from this specific *Ureaplasma* species and serotype is identical to that of other species and serotypes and thus is a model for all *Ureaplasma* species, this should be explained for clarity. If not, then limitations of this model must be expounded upon further.

Thank you for this important point which we have addressed in the revised manuscript.

3) A final concern regards the use of *E. coli* expressing *Ureaplasma* urease gene clusters in the model. This is an adequate model to isolate the contribution of the specific urease gene cluster expressed on this plasmid. However, no comment is made as to why an artificial expression model was used rather than live *U. urealyticum* and/or *U. parvum* (as was used in references 11 and 12). This needs to be elucidated in the manuscript, with limitations of this model clearly explained.

Thank you for raising this question. The *E. coli* model was used primarily to show that the ammonia production witnessed was a result of urease expression alone, and not infection/lack of infection. Additionally, *E. coli* is more predictable and easier to manage in the *in vivo* experiments. We have discussed these issues in the revised manuscript.

Additional comments are as follows:

- The schematics presented in the introduction (Figures 1 and 2) seem better suited for a review of hyperammonemia. They are well designed and visually impressive, but the specific mechanisms presented (particularly those in Figure 1 involving ammonia and glutamate metabolism in astrocytes and neurons) lead the reader to think these will be experimentally targeted in the body of the work. They are not, and the reader is left wondering why such detail was included that does not contextualize the specific experiments presented.

We agree that the mechanisms of brain toxicity are better suited to a review and have removed the figure. The other figures have been renumbered accordingly. However, given that this is a condition not well-known in the field of microbiology, and that the proposed mechanisms are unique, we feel that figure 2 (now figure 5) is useful in conveying the proposed pathophysiology. However, we have moved the figure to the discussion, after acute renal failure is mentioned as a possible cause of uremia.

- A hypothesis is presented in the abstract and the discussion, but not in the introduction. Please include hypotheses in the introduction to provide the reader with an understanding of the questions addressed in the study.

We apologize for this oversight. A hypothesis has been added to the end of the introduction, as follows.

- Even when non-significant p values are present, please include the p values in the text rather than stating that compared values are "not significantly different".

Thank you. Given that there were many comparisons made, all p-values have now been supplied in the supplementary figures referenced in the text, rather than the text itself.

- Data reporting the BUN of experimental animals is repeated three times in the manuscript (lines 123-124, 217-218, and 262). It is sufficient to only report this in the results and then comment in the discussion.

We apologize for the repetitiveness. We have left the mention on lines 123-124 of the original manuscript as the only instance the numbers are listed.

- There are minor typos throughout the manuscript, the most significant being the title subheading on line 211-212 describes an "E. coli strain expressing U. urealyticum". It seems the term "urease gene cluster" was omitted.

Thank you. The manuscript has been thoroughly reviewed and edited for spelling and grammar.

Reviewer #2 (Comments for the Author):

- Please clarify why you are using CCU as a quantification measure, instead of CFU. CFU is by far the more accurate measure of growth, so it would be useful to provide a rationale for CCU which of course can be used, but is not as accurate.

CCU was used for high throughputness of the assay, and to avoid population loss that may occur in alkaline, high-titer *Ureaplasma* culture in which the organisms dry out on solid media. The CCU findings are in line with the qPCR counts from the experiments of figure 3.

- Lines 89-96: did you assess ammonia production from the urease-containing plasmid prior to inoculation into animals?

We apologize for not qualifying in the text that ammonia production was confirmed via inoculation onto urea slants prior to inoculation. We have added text to the Methods section

- With regards to your animal challenge methodology - 50 uL seems like a large volume for intratracheal challenge. I suspect this would overflow and some of the volume would go into the stomach. Can you please comment on the volume used here as I think a smaller volume would have been better for this particular challenge route in the mice, to ensure the total volume remained in the required site. Similarly, can you please clarify for each experiment how many

mice were used for each experimental group and the statistical tests used to inform the appropriate experimental group size.

We apologize for the oversight in animal numbers. The numbers were 12 mice per group (6 females and 6 males). Prior to experimentation, we had calculated that with a sample size of at least 10 mice per group, we would have at least 80% power to detect a difference of 1.33 standard deviations or larger for levels of ammonia between two groups using two sample t-tests. The number of animals has been added to the Methods section as well as the legend of figure 4.

50 μ L instillation was used to ensure that enough volume (and cells) to infect the lungs, not just the upper airway. We utilized this method in the past, and though it is possible that all 50 μ L may not enter the airways, it was pertinent to ensure that a significant amount of high-titre culture was introduced. A justification has been added to the Methods section.

- in your dialysed flow system, I can see that media is added and spent media removed at 2 ml per hour. Did you also take into account the need for additional volumes of media to replace the volumes of media being taken for sampling (or was the spent media used to assess the urea/ammonia concentrations? Some clarification on this point would be good as changes in the total volume of the system can affect the concentrations of some of these substrates.

This is an important point. Sampling was done from the exterior of the dialysis tube, however the volume removed was a very small percentage of the total (0.04% per draw, or .2 mL from 250 ml) that we do not feel would significantly impact the results. However, we have added this as a limitation of the study.

- For your nucleic acid extraction, can you please clarify if you have used the manufacturer's instructions or if there was any deviation from the standard workflow/instructions.

Thank you for catching this. Nucleic acid extraction was performed per the manufacturer's instructions. This has been clarified in the Methods section.

- For figure 4, please change the black bars (or the bottom error bar) so you can clearly see the variation in results. Also note in the legend for this figure, you currently have written " **p {less than or equal to}0.01, **p {less than or equal to}0.0001" and this needs to be amended (presumably the second value is meant to be for ****?)

Thank you for the suggestion; we apologize for the error in the legend which has been corrected.

- I think you could be a little more transparent with the limitations of this work and this could be addressed with a few additional sentences added to the discussion.

We have added additional limitations to the Discussion.

January 16, 2022

Dr. Robin Patel
Mayo Clinic
200 First Street SW
Rochester, Minnesota 55905

Re: Spectrum01942-21R1 (Contribution of Uremia to *Ureaplasma*-Induced Hyperammonemia)

Dear Dr. Robin Patel:

Your manuscript has been accepted, and I am forwarding it to the ASM Journals Department for publication. You will be notified when your proofs are ready to be viewed.

Sincerely,

Karen Carroll
Editor, Microbiology Spectrum
